# Transcriptional signatures in human macrophage-like cells infected by *Leishmania infantum, Leishmania major* and *Leishmania tropica*

**Aurora Diotallevi**[1], **Federica Bruno**[2], **Germano Castelli**[2], **Giuseppe Persico**[3], **Gloria Buffi**[1], **Marcello Ceccarelli**[1], **Daniela Ligi**[1], **Ferdinando Mannello**[1], **Fabrizio Vitale**[2], **Mauro Magnani**[1], **Luca Galluzzi**[1]*

1 Department of Biomolecular Sciences, University of Urbino Carlo Bo, Urbino, Italy, 2 Centro di Referenza Nazionale per le Leishmaniosi (C.Re.Na.L.), OIE Leishmania Reference Laboratory, Istituto Zooprofilattico Sperimentale della Sicilia A Mirri, Palermo, Italy, 3 Department of Experimental Oncology, IRCCS, European Institute of Oncology, Milan, Italy

* luca.galluzzi@uniurb.it

## Abstract

### Background

In the Mediterranean basin, three *Leishmania* species have been identified: *L. infantum*, *L. major* and *L. tropica*, causing zoonotic visceral leishmaniasis (VL), zoonotic cutaneous leishmaniasis (CL) and anthroponotic CL, respectively. Despite animal models and genomic/transcriptomic studies provided important insights, the pathogenic determinants modulating the development of VL and CL are still poorly understood. This work aimed to identify host transcriptional signatures shared by cells infected with *L. infantum*, *L. major*, and *L. tropica*, as well as specific transcriptional signatures elicited by parasites causing VL (i.e., *L. infantum*) and parasites involved in CL (i.e., *L. major*, *L. tropica*).

### Methodology/Principal findings

U937 cells differentiated into macrophage-like cells were infected with *L. infantum*, *L. major* and *L. tropica* for 24h and 48h, and total RNA was extracted. RNA sequencing, performed on an Illumina NovaSeq 6000 platform, was used to evaluate the transcriptional signatures of infected cells with respect to non-infected cells at both time points. The EdgeR package was used to identify differentially expressed genes (fold change > 2 and FDR-adjusted p-values < 0.05). Then, functional enrichment analysis was employed to identify the enriched ontology terms in which these genes are involved. At 24h post-infection, a common signature of 463 dysregulated genes shared among all infection conditions was recognized, while at 48h post-infection the common signature was reduced to 120 genes. Aside from a common transcriptional response, we evidenced different upregulated functional pathways characterizing *L. infantum*-infected cells, such as VEGFA-VEGFR2 and NFE2L2-related pathways, indicating vascular remodeling and reduction of oxidative stress as potentially important factors for visceralization.

**Data Availability Statement:** The raw fastq files were deposited in Sequence Read Archive (SRA)

(BioProject ID: PRJNA995506) with accession numbers SAMN36470832-SAMN36470847.

**Funding:** This work was supported by Ricerca Corrente 2018 (IZS SI 02/18) from the Italian Ministry of health (awarded to FV), and by FANOATENEO (awarded to MM). The funders had no role in study design, data collection and analysis, decision to publish, or preparation of the manuscript.

**Competing interests:** The authors have declared that no competing interests exist.

## Conclusions

The identification of pathways elicited by parasites causing VL or CL could lead to new therapeutic strategies for leishmaniasis, combining the canonical anti-leishmania compounds with host-directed therapy.

## Author summary

Leishmaniasis is a zoonosis caused by the intracellular parasite of the genus *Leishmania*. In the Mediterranean region, the human disease can be present mainly in two forms: cutaneous leishmaniasis (CL) and visceral leishmaniasis (VL), which can be fatal if untreated. The clinical presentation of the disease depends on the parasite species and host characteristics. However, the molecular mechanisms underlying the development of the visceral form of leishmaniasis are not yet fully understood. In this study, we have determined the gene expression profile in cells infected by three *Leishmania* species (*L. infantum*, *L. major*, *L. tropica*) responsible for different forms of the disease. The results have allowed us to identify an expression profile (and the related pathways) shared by cells infected by all *Leishmania* species. Aside this common response, we also identified a transcriptional response distinguishing cells infected by *L. infantum*, *L. major* or *L. tropica*, as well as the relevant pathways in which these genes participate. Therefore, our results contribute to the knowledge of the host cell mechanisms that are activated along infection with different *Leishmania* species causing different forms of the disease and, therefore, may help develop new host-directed therapy for Leishmaniasis.

## Introduction

Leishmaniasis is a neglected tropical disease, diffuse in both the Old World (Europe, Africa, Asia) and New World (the Americas) [1] occurring in three main forms, depending on the *Leishmania* species and host characteristics: cutaneous leishmaniasis (CL), mucocutaneous leishmaniasis (MCL) and visceral leishmaniasis (VL, also known as kala-azar), the most severe form with important symptoms (e.g., hepatosplenomegaly, high fever, pancytopenia, hypergammaglobulinemia) that may lead to death if untreated [2]. Visceral leishmaniasis, causing 20,000 to 40,000 deaths per year [1], is generally caused by *L. donovani* and *L. infantum* species. In the Mediterranean basin, three *Leishmania* species have been identified: *L. infantum*, causing mainly zoonotic visceral leishmaniasis; *L. major*, causing zoonotic (wet) cutaneous leishmaniasis; and *L. tropica*, causing anthroponotic (dry) cutaneous leishmaniasis [3].

The determinants of different forms of leishmaniasis are still poorly understood, although animal models and genomic/transcriptomics studies are beginning to provide important insight into this disease. Several factors, such as vector, host, and pathogen determinants, can mediate the development of visceral and cutaneous leishmaniasis. Identifying those factors is highly relevant to understand disease progression, parasite survival mechanisms and potentially to develop better therapies/vaccines or to identify epidemiologic markers for virulence.

Transcriptomic analyses through microarrays and RNA-Seq have been implemented in infected macrophages (the main resident cell for *Leishmania* parasites) to understand how host response can be modulated by different *Leishmania* species [4–9]. Most of these studies have been performed in murine models. Overall, the results suggest that the modulation of

gene expression in macrophages is dynamic and could be linked to the *Leishmania* infecting species as well as to the time of infection.

For instance, it has been shown that *L. donovani* and *L. major* induced remarkably similar gene expression profiles in murine bone marrow-derived macrophages at 24h post-infection, although *L. donovani* induced higher expression levels of some genes (e.g., Ptgs2/COX2, prostaglandin E synthase, matrix metalloprotease 13, metallothionein 1 and 2) [9]. Other studies have shown that the host's genetic background influences disease development [10]. In particular, NRAMP1/SLC11A1 (a transporter involved in iron metabolism and host response to intracellular parasites) and a number of cytokines and their receptors (e.g., TNF-α, IL-4, TGF-β, IL-2 receptor; CXCR-2) have been associated with susceptibility to visceral disease [11].

Changes in gene expression upon *Leishmania* infection have been documented in various human and murine cell models, explaining different aspects of host response modulation elicited by the parasite [12]. However, a full comprehension of the mechanism of pathogenesis has yet to be reached, as well as determinants leading to VL in hosts infected with viscerotropic species. Moreover, to the best of our knowledge, there are no data that systematically compares the transcriptomic profile of human macrophages infected with *Leishmania* species causing different disease outcomes in Mediterranean basin (i.e. *L. infantum*, *L. major*, or *L. tropica).*

In this work, we investigated the global gene expression in a human macrophage-like cell line (U937 cells) in response to infection with *L. infantum*, *L. major* and *L. tropica* to identify shared and/or specific transcriptional signatures driven by parasites potentially causing visceral and cutaneous disease.

## Methods

### Parasite culture

*L. infantum* (MHOM/IT/08/31U), *L. major* (MHOM/SU/73/ASKH) and *L. tropica* (MHOM/SU/74/K27) taken from strain collection of the WOAH Leishmania Reference Laboratory (National Reference Centre for Leishmaniosis, Palermo, Italy), were cultivated in RPMI-PY medium supplemented with 10% heat-inactivated Fetal Bovine Serum (FBS), 1% glutamine and antibiotic solution (250 μg/ml gentamicin and 500 μg/ml 5-fluorocytosine) [13,14]. Cultures were monitored every two days for the presence of flagellated promastigotes under the microscope. Stationary growth parasites were used in the infection experiments.

### Cell culture-derived macrophages and infection

The human monocytic cell line U937 (ATCC CRL-1593.2) was cultured in RPMI-1640 medium supplemented with 10% Fetal Bovine Serum (FBS), 2 mM L-glutamine, 1% penicillin/streptomycin at 37°C and 5% $CO_2$. The U937 cells in the logarithmic phase of growth were plated at the concentration of $6 \times 10^5$ cells in 35 mm dishes containing 25 ng/mL of phorbol 12-myristate 13-acetate (PMA) for 18 h to induce macrophage differentiation [15,16]. All cell culture reagents were purchased from Merck. The three *Leishmania* reference strains stationary promastigotes were used to infect U937-derived macrophages with a parasite-to-cell ratio of 5:1, according to Bruno et al. [16]. Non-infected cells treated with PMA were used as control. Each infection was repeated twice (i.e., two biological replicates). After 24 h and 48 h, cells were washed to remove free parasites and directly lysed for downstream analyses. The infection index was calculated in cells stained with 100 mg/mL ethidium bromide by multiplying the percentage of infected macrophages by the average number of parasites per macrophage. At least 200 macrophages were counted by visual examination at 40X magnification, using a Leica DM4000 fluorescence microscope (Leica, Wetzlar, Germany) to determine the number of resident amastigotes.

## RNA extraction

Macrophage-like cells were directly lysed with 700 μl of QIAzol Lysis Reagent (Qiagen, Hilden, Germany). Total RNA extraction was performed with the miRNeasy Mini Kit (Qiagen, Hilden, Germany) following the manufacturer's instructions. To quantify extracted RNA, the Qubit 4 Fluorometer and the RNA HS Assay (Thermo Fisher Scientific, Waltham, MA, USA) were used. The RNA integrity/quality was preliminarily assessed by 1.2% agarose gel stained with Midori green (Nippon Genetics, Europe). For RNA sequencing, the RNA integrity number (RIN) was determined using an Agilent 2100 Bioanalyzer (Agilent Technologies, Inc.). Samples with RIN > 9.0 were used for library construction.

## RNA sequencing

Library preparation was performed from 1 μg of total RNA using the TruSeq Stranded mRNA Library Prep Kit (Illumina, San Diego, CA, USA), following the manufacturer's protocol. The prepared libraries were assessed for size distribution using the Agilent 2100 BioAnalyzer and a DNA 1000 chip and were quantified using qPCR according to the Illumina qPCR Quantification Protocol Guide. RNA libraries were sequenced as 100 bp paired-end runs on an Illumina NovaSeq 6000 platform. Library preparation and sequencing were performed by Macrogen Europe NGS service (Amsterdam, the Netherlands).

## RNA-seq analysis

Quality control of raw sequencing data was done using FastQC tool. Mapping to a human reference genome assembly (UCSC hg38) was done using STAR [17]. After removal of low expressed genes (only genes with CPM (count per million) < 1 in at least 1 sample), the remaining expressed genes (n = 13,093) were used for differential gene expression analysis using EdgeR package [18]. The TMM (Trimmed Mean of M-values) normalization was applied on row count while the exact test edgeR approach was used to make pairwise comparisons between groups, followed by Benjamini-Hochberg approach to calculate the FDR. Genes with an absolute fold change > 2 and FDR-adjusted p-values < 0.05 were identified as differentially expressed. The raw fastq files were deposited in Sequence Read Archive (SRA) (BioProject ID: PRJNA995506) with accession numbers SAMN36470832-SAMN36470847. Graphs and Venn diagrams were drawn using Microsoft excel and the webtool available at https://bioinformatics.psb.ugent.be/webtools/Venn/, respectively.

## Functional enrichment

To identify ontology terms dysregulated by infection, a functional enrichment analysis was performed using the multiple gene list tool in Metascape (http://metascape.org) [19]. A multiple gene list is a table containing three lists of deregulated genes, one for each infecting *Leishmania* species. In this case, pathway enrichment analysis is applied to all individual gene lists independently and results are merged and clustered to identify pathways shared across all gene lists. The enrichment analysis was performed separately for up- and downregulated genes [20] and was carried out with the following ontology sources: KEGG Pathway, Reactome Gene Sets, WikiPathways. All genes in the human genome have been used as the enrichment background. Significant functional enrichment terms were defined as those with a p-value < 0.001, a minimum count of 3, and an enrichment factor > 1.5.

**Table 1. Transcripts and primers used in qPCR.**

| Target mRNA (NCBI Ref Seq) | Forward Primer | Reverse Primer | Reference |
|---|---|---|---|
| HPRT (NM_000194.3) | TATGCTGAGGATTTGGAAAGGG | AGAGGGCTACAATGTGATGG | [22] |
| GAPDH (NM_002046.7) | CCATGTTCGTCATGGGTGTG | GGTGCTAAGCAGTTGGTGGTG | [23] |
| IFIT1 (NM_001548.5) | ATGGGCCTTGCTGAAGTGTG | TCAGGGTTTTCAGGGTCCAC | |
| IFI6 (NM_002038.3) | TCTGCGATCCTGAATGGGG | TATTACCTATGACGACGCTGCT | |
| MMP3 (NM_002422.5) | ATGATGATGAACAATGGACAAAGGA | TTGGCTGAGTGAAAGAGACCC | |
| OAS2 (NM_001032731.2) | GACACTGATCGACGAGATGGT | TACCATCGGAGTTGCCTCTT | [24] |
| JUN (NM_002228.4) | GAGCTGGAGCGCCTGATAAT | CCCTCCTGCTCATCTGTCAC | [25] |
| DHCR7 (NM_001360.3) | CAGCGCCAGAGACTGCAAAT | CGCCCATTGAAGAACAGCTT | |
| HMGCS1 (NM_001098272.3) | GCACAGAAGAACTTACGCTCG | GGGCTTGGAATATGCTCAGTTG | |
| TMEM97 (NM_014573.3) | ATCCCCATCACCCTGTTCAT | GTCTTTGAACTCCTTAGCATACCAC | |
| STAT1 (NM_001384880.1) | TGGCACCAGAACGAATGAGG | CTGGCTGACGTTGGAGATCA | |
| IL1B (NM_000576.3) | CAAACCTCTTCGAGGCACAA | GGCTGCTTCAGACACTTGAG | |
| MMP10 (NM_002425.3) | CCTTGTGCTGTTGTGTCTGC | ACTTTTCTAGGTATTGCTGGGCA | |
| HMOX1 (NM_002133.3) | AAGACTGCGTTCCTGCTCAA | GGTCCTTGGTGTCATGGGTC | |
| IL6 (NM_000600.5) | GTGAAAGCAGCAAAGAGGCAC | GATTTTCACCAGGCAAGTCTCC | |
| HSPA5 (NM_005347.5) | CCCCGAGAACACGGTCTTT | CAACCACCTTGAACGGCAA | [23] |
| ACAT2 (NM_005891.3) | CTGGGCTCCACTGTCATCAA | ACCCACACTGGCTTGTCTAA | |
| MYC (NM_002467.6) | CGTCCTCGGATTCTCTGCTC | TTGTTCCTCCTCAGAGTCGC | |
| HMGCR (NM_000859.3) | TCCCAGCTTGTGTGTCCTTG | AAACTCGGGCAAAATGGCTG | |
| MMP9 (NM_004994.3) | CAGTCCACCCTTGTGCTCTT | CCCGAGTGTAACCATAGCGG | |

## RNA-seq validation through qPCR

RT-qPCR validation was performed on 18 dysregulated genes (Table 1) using total RNA samples subjected to RNA-seq. Reverse transcription was performed using 500 ng total RNA using PrimeScript RT Master Mix (Perfect Real Time) (Takara Bio Inc.) according to the manufacturer's instructions. Equal amounts of cDNA (1 μl) were assessed in total volumes of 20 μl containing TB Green Premix Ex Taq II (Tli RNaseH Plus) Mastermix (Takara Bio Europe, France) and primers (200 nM) (Table 1). The mixtures were incubated at 95˚C for 10 min, followed by 40 cycles at 95˚C for 10 s, and 60˚C for 50 s. The reactions were carried out in duplicate using a Rotorgene Q (Qiagen). The fold changes were calculated by relative quantification

using the ΔΔCt method [21]. The data were normalized using HPRT and GAPDH as reference genes, and the relative gene expression was set to 1 for the control (non-infected) samples.

## Western blotting

Infection was repeated as described above. U937-derived macrophages were processed for Western blot analysis as previously reported [26]. Briefly, cells were directly lysed 20 min in ice with 20 mmol/L Hepes pH 7.9, 25% v/v glycerol, 0.42 mol/L NaCl, 1.5 mmol/L MgCl2, 0.2 mmol/L EDTA, 0.5% v/v Nonidet P-40, 1 mmol/L DTT, 1 mmol/L Naf, 1 mmol/L Na3VO4, and 1X complete protease inhibitor cocktail (Roche Diagnostics Ltd.,Mannheim, Germany). Samples were then frozen and thawed twice and clarified by centrifugation at 12,000 rpm for 10 min at 4˚C. Total cell lysates were fractionated by SDS-PAGE, and gels were electroblotted onto a nitrocellulose membrane (0.2 μm pore size) (Bio-Rad Laboratories, Inc., Hercules, CA, USA). The resulting blots were probed with the following primary antibodies: anti-CHOP/ DDIT3 (#2895), anti-ATG7 (#2631), anti-NQO1 (#3187) purchased from Cell Signaling Technology (Beverly, MA, USA); anti-c-Myc (ab32072) purchased from Abcam (Toronto, ON, Canada, C); anti-GRP78/HSPA5 (sc-166490), anti- Ero1-L-α (sc-365526) purchased from Santa Cruz Biotechnology Inc. (Santa Cruz, CA, USA); anti-actin (A2066) purchased from Sigma-Aldrich. Signals were detected using horseradish peroxidase-conjugated secondary antibodies and blots were treated with chemiluminescence reagents (Clarity Western ECL Substrate) (Bio-Rad Laboratories, Inc., Hercules, CA, USA). The immunoreactive bands were detected and quantified by Chemi-Doc System (Bio-Rad Laboratories, Inc., Hercules, CA, USA) equipped with Image Lab 6.1 software.

## Elisa test

The concentration of IL-1β was determined in cell culture supernatants of U937-derived macrophages infected 24h and 48h with *L. infantum* using the sandwich ELISA assay Human IL-1β ELISA Kit (Sigma), following the manufacturer's protocol. The absorbance was read using a microplate reader Spectrostar Nano (BMG Lab Tech) at 450 nm immediately.

## Gelatin zymography

U937-derived macrophages non-infected and infected by *L. infantum* at 24h and 48h (using FBS-free medium) were directly lysed with ice-cold 0.1% NP40-PBS and spinned 10 sec. The lysed supernatants were quantified using Bradford method and 16.6 μg of total proteins were loaded in each lane. Gelatin zymography was carried out with fixed-concentration (7.5%) polyacrylamide separating gels copolymerized with 3 g/L 90 Bloom Type A gelatin and 4% stacking gel as described previously [27]. Briefly, zymogram gels were run in cold SDS running buffer (25 mM Tris, 192 mM glycine, and 0.1% w/v SDS) at a constant voltage of 120 V, for about 2 h. After electrophoresis, gels were washed twice for 20 min at room temperature in Triton X-100 2.5% to gently remove SDS. Gels were washed with distilled water and incubated for 22 h at 37˚C in an Enzyme Incubation Buffer (EIB) containing 50 mM Tris, 5 mM CaCl$_2$, 100 mM NaCl, 1 mM ZnCl$_2$, 0.3 mM NaN$_3$, 0.2 g/L of Brij-35, and 2.5% of Triton X-100, pH 7.6. Staining was performed using Coomassie Brilliant Blue R-250 (0.2% w/v Coomassie in 50% methanol and 20% acetic acid). Gels were maintained in destaining solution (50% v/v methanol and 20% v/v acetic acid) until clear gelatinolytic bands appeared against the uniform dark-blue background of undigested protein substrate.

### Statistical analysis

Concordance between the RNA-seq and RT-qPCR data or protein levels was evaluated by Spearman's correlation analysis, carried out using GraphPad Prism 8 (GraphPad Software, Inc., La Jolla, CA, USA).

## Results and discussion

### Identification of differentially expressed genes upon *Leishmania* infection

To compare the global transcriptional response, the transcriptomes of U937-derived macrophages infected by *L. infantum*, *L. major* and *L. tropica* at 24 h and 48 h were simultaneously profiled using RNA-seq. At both time points, the infection was well established for all *Leishmania* species (S1 Fig). Over 30 x $10^6$ reads per sample were identified (average 31.3 x $10^6$). For infected macrophages, the trimmed reads mapped on the human genome were in average 44.6 ± 3.0% and 66.1 ± 3.0% at 24 h and 48 h, respectively, while for the uninfected macrophages were 98.5 ± 0.1% and 98.8 ± 0.4% at 24 h and 48 h, respectively (values are mean ± SEM). The lower percentage values obtained from infected samples were due to reads coming from parasite-derived transcripts, as demonstrated by the fact that in average 43.9 ± 6.7% and 26.9 ± 4.9% of trimmed reads mapped on *Leishmania* genomes at 24 h and 48 h, respectively. The most abundant transcripts (>91.3%) were protein-coding RNA. Overall, the reads mapped to 15,013 protein-coding genes on the UCSC hg38 human genome. From that, at 24h post-infection, we identified 749 up- and 864 downregulated genes in *L. infantum*-infected cells, 401 up- and 500 downregulated genes in *L. major*-infected cells, and 706 up- and 733 downregulated genes in *L. tropica*-infected cells. At 48h post-infection, we identified 124 up- and 330 downregulated genes in *L. infantum*-infected cells, 116 up- and 353 downregulated genes in *L. major*-infected cells, and 54 up- and 132 downregulated genes in *L. tropica*-infected cells (Fig 1). The complete lists of genes are provided in S1 Table (24h) and S2 Table (48h). A direct comparison of deregulated genes at each timepoint revealed that the macrophage response to infection at 24 h was more pronounced compared to the response at 48 h, confirming previous findings in murine macrophage infection models, which evidenced a greater gene expression modulation during early infection and a progressive reduction at later timepoints until 72 h post-infection [4,6,28,29]. A fraction of these genes was consistently up- or downregulated both at 24 h and 48h, while another fraction (19% to 69%) was deregulated uniquely at 48h post-infection (Tables 2 and S3).

### Validation of differential gene expression

RNA-seq results were validated by RT-qPCR on 18 differentially regulated genes (IFIT1, IFI6, JUN, MMP3, OAS2, STAT1, IL1B, MMP10, HMOX1, IL6, HSPA5, DHCR7, HMGCS1, TMEM97, ACAT2, MYC, HMGCR, MMP9) in each infection condition, using the same RNA samples. Comparative analyses showed high concordance between the RNA-seq and RT-qPCR data either at 24h (Spearman's rho, $\rho$ = 0.98, p<0.001) and 48h (Spearman's rho, $\rho$ = 0.90, p<0.001) (Fig 2), thus validating the RNA-seq results.

To further strengthen RNA-seq findings, several dysregulated genes were monitored at the protein level under the same experimental conditions. For instance, DDIT3, ATG7, ERO1-L-α (ERO1A), NQO1, HSP70 family protein 5 (HSPA5), and c-Myc (MYC) were monitored in all infection conditions by western blotting, while IL-1β and MMP-2/MMP-9 were monitored in *L. infantum*-infected cells by ELISA test and gelatin zymography, respectively. Overall, the results were in agreement with the dysregulation found at the mRNA level (S2 Fig). In particular, the correlation between gene expression and protein levels was evident at 24h post-infection (Spearman's rho, $\rho$ = 0.87, p<0.001), while it was not significant at 48h post-infection,

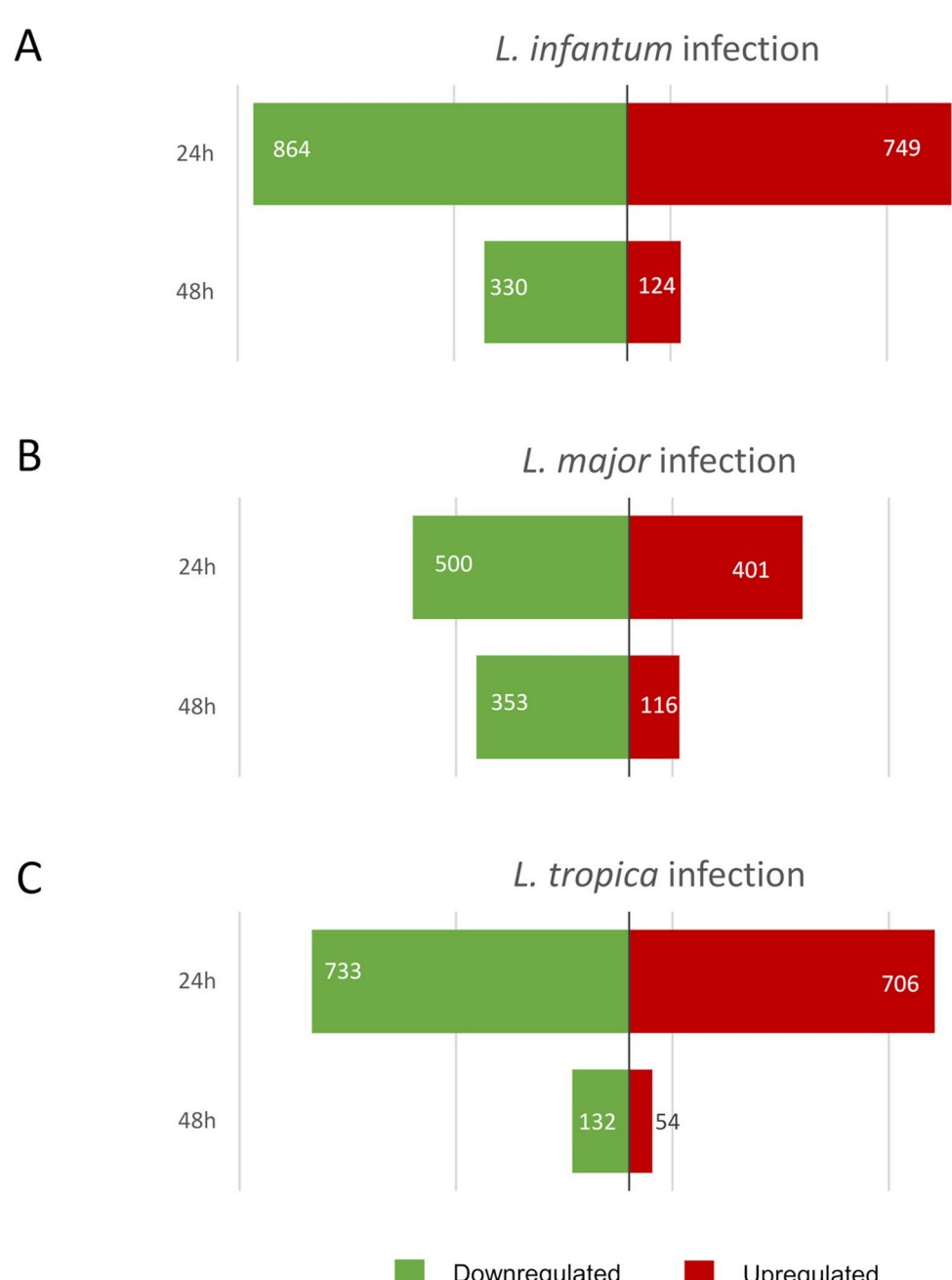

**Fig 1. Differentially expressed genes in macrophages infected with *Leishmania* species.** The numbers of upregulated (right, red) and downregulated (left, green) genes in A) *L. infantum*-, B) *L. major*-, and C) *L. tropica*-infected macrophages relative to uninfected controls are depicted as horizontal bar plots at 24 and 48 h post-infection. The complete lists of dysregulated genes are provided in S1 and S2 Tables.

when the expression of all considered genes (with the exception of ATG7) did not change significantly.

## Defining the expression signatures upon *Leishmania* infection

Despite the similarities among parasites of the subgenus *Leishmania* (*Leishmania*), the different species present different patterns of virulence and pathologies. For instance, *L. infantum*

**Table 2. Number of unique and common genes deregulated at 24h and 48h post-infection.**

| | | Unique genes 24h (%) | Unique genes 48h (%) | Common genes between 24 h and 48 h |
|---|---|---|---|---|
| *L. infantum*-infected cells | Upregulated | 659 (88%) | 34 (27%) | 90 |
| | Downregulated | 643 (74%) | 109 (33%) | 221 |
| *L. major*-infected cells | Upregulated | 322 (80%) | 37 (32%) | 79 |
| | Downregulated | 390 (78%) | 243 (69%) | 110 |
| *L. tropica*-infected cells | Upregulated | 662 (94%) | 10 (19%) | 44 |
| | Downregulated | 650 (89%) | 49 (37%) | 83 |

(as well as *L. donovani*) is considered the causative agent of VL, while *L. major* and *L. tropica* are considered etiological agents of CL. These differences depend on the immunological status of the host but also on species determinants that can induce different responses in host cells. Some of these determinants have been investigated in the past years; however, there are no data that systematically compare the transcriptomic profile of macrophages infected with *Leishmania* species causing different disease outcomes in old world.

We explored whether *L. infantum*, *L. major* and *L. tropica* elicited significantly different responses in U937-derived macrophages at 24h and 48h post-infection. The RNA-seq and Venn diagram analysis revealed a response shared by macrophages infected with all species but also some significantly different response depending on the *Leishmania* species (Fig 3). Notably, at 24h post-infection, we recognized a common signature of 463 genes (237 up and 226 downregulated) shared among all infection conditions, while at 48h post-infection the common signature was reduced to 120 genes (38 up and 82 downregulated). All shared and unshared genes are listed in S4 Table.

At 24h post-infection, among the common upregulated genes, most recognizable were several interferon signaling-related genes and genes involved in response to cytokines (e.g., IFIT2, IFIT1, IFIT3, IFIT5, IRF7 [30], MX2, OAS1, OAS2, OAS3, OASL, EIF2AK2, STAT1, ISG15, TRIM14, TRIM22, TRIM34, TRIM5 USP18, XAF1, SAMHD1, HERC5, RSAD2, GBP5, TNFSF13B), as well as genes involved in protein ubiquitination (e.g., NEDD4L, DTX3L, TRIM38, HERC5, HERC6). Moreover, several genes encoding metallothionein 1 family members (E F G X), previously shown to be upregulated in macrophages infected with *Leishmania* [6,31], were found, even though their potential role in establishing infection has not been clarified yet. Other common upregulated genes included: HMOX1 (an antioxidative stress and anti-inflammatory gene regulated by NFE2L2/NRF2, playing also a role in iron homeostasis) [32], JUN (coding for a transcription factor that heterodimerizes with proteins of the FOS family and binds to the AP-1 consensus motif), PARP9 and PARP14 that have opposing roles in macrophage activation. PARP9 positively regulates pro-inflammatory cytokines production in response to IFNG stimulation by suppressing PARP14-mediated STAT1 ADP-ribosylation and thus promoting STAT1 phosphorylation [33]. Among the common downregulated genes, many were involved in cholesterol biosynthesis (i.e., ACAT2, DHCR7, DHCR24, FDFT1, FDPS, HMGCR, HMGCS1, LSS, MVD, MVK, MSMO1, SQLE, SREBF2, EBP, NSDHL) and in the biosynthesis of saturated and unsaturated fatty acids (FASN and FADS2, respectively). Moreover, APOA4 (a potent activator of lecithin-cholesterol acyltransferase in vitro) was also present. Other genes included transporters such as SLC25A3 (mitochondrial phosphate transporter), SLC43A2 (neutral amino acids transporter) and SLC26A11 (sodium independent sulfate transporter), and the interleukin receptors IL10RA (that participates in IL10-mediated anti-inflammatory functions) and the IL22 antagonist IL22RA2.

At 48h post-infection, among the common upregulated genes, several interferon signaling-related genes, genes involved in response to cytokines (e.g., IFIT1, IFIT3, IFIT5, MX2, OAS2,

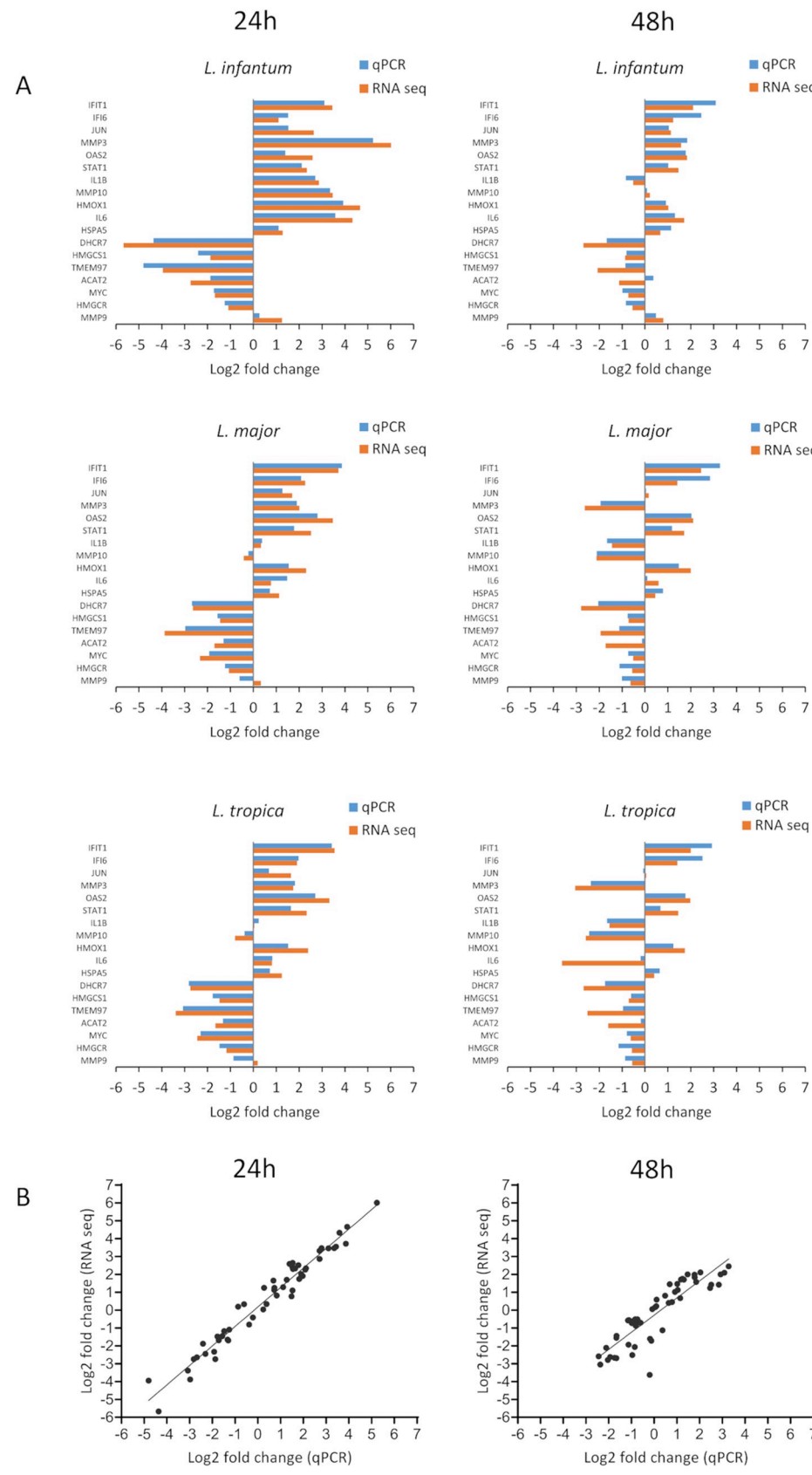

**Fig 2. qPCR validation of RNA-seq data.** A) Expression profiles of selected genes obtained by RNA-seq and qPCR in cells infected by *L. infantum*, *L. major* and *L. tropica*, both at 24h and 48h post-infection. Expression was normalized using GAPDH and HPRT as reference genes. Results represent mean values and are expressed as Log2 values of the fold change. B) Correlation analysis of RNA-seq and qPCR results, at 24h post-infection (Spearman's rho, ρ = 0.98, p<0.001) and 48h post-infection (Spearman's rho, ρ = 0.90, p<0.001).

OAS3, EIF2AK2, STAT1, TRIM14, TRIM22, TRIM34, USP18, XAF1, TNFSF13B) as well as PARP9 and PARP14, were still upregulated. Among the common downregulated genes, many genes involved in cholesterol biosynthesis (i.e. DHCR7, FDFT1, FDPS, MVD, MSMO1, SREBF2, EBP) and in the biosynthesis of unsaturated fatty acids (FADS2) were still downregulated. The chemokine CCL5, which was downregulated only in *L. tropica*-infected cells at 24h, appeared downregulated in all infection conditions at 48h.

Besides the common expression signature, the unique sets of genes deregulated in *L. infantum*-, *L. major*- and *L. tropica*-infected cells (Fig 3 and S4 Table) could provide information about the pathways related to specificities of clinical manifestations. In particular, the deregulated genes in *L. infantum*-infected cells could indicate the determinants of VL and some possible drug targets for the most severe form of disease.

At 24h post-infection, among the 398 genes upregulated exclusively in *L. infantum*-infected cells were the cytokines IL1B, IL1A, IL6 the IL12 receptor subunit β1 (IL12RB1), the immuno-modulator CSF2, the transcription factor NFE2L2 (that plays a key role in the response to oxidative stress), GCLC and GCLM (codifying for two subunit of glutamate–cysteine ligase, a key enzyme involved in glutathione synthesis) and SLC7A11 (coding for a transporter of cystine and glutamate and involved in GSH production) [34,35]. Also, several genes coding for matrix metalloproteases (i.e., MMP3, MMP9, MMP10, MMP12, MMP13, MMP19) were found to be upregulated. These enzymes are important in many physiological as well as pathological conditions; in fact, they not only can degrade extracellular matrix molecules (e.g., collagen, laminin, fibronectin) but also can act on other molecules such as cytokines, hormones and chemokines,

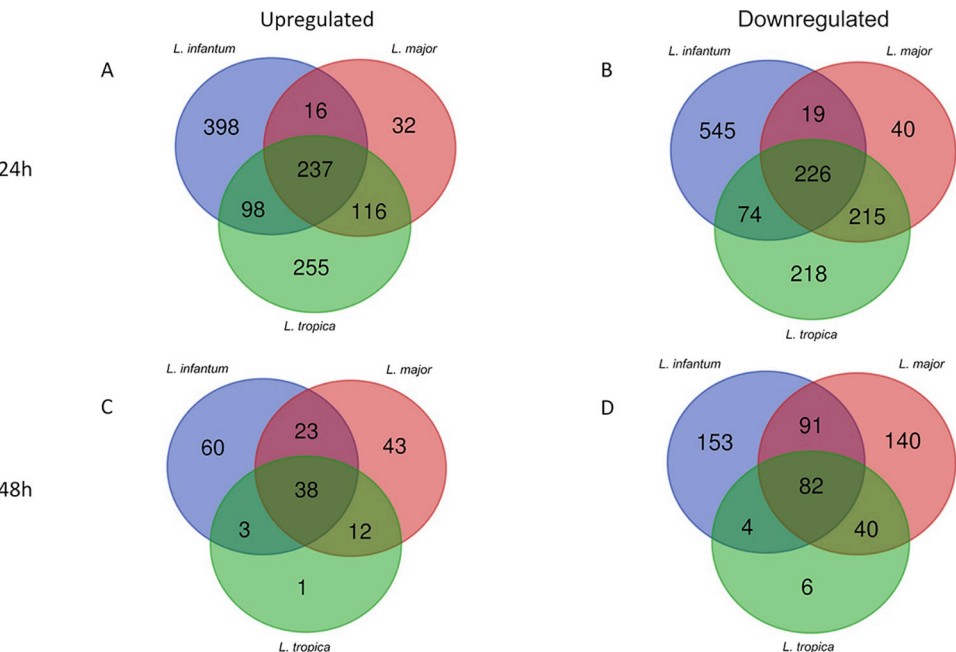

**Fig 3. Venn diagram analysis of DEGs in U937-derived macrophages infected by *L. infantum* (blue), *L. major* (red) and *L. tropica* (green).** Venn diagram of the genes upregulated (A, C) and downregulated (B, D) in response to infection after 24 h and 48 h, showing the numbers of unique and common genes for each comparison.

thereby regulating immune response. For example, MMP-9 cleaves and potentiates IL-8 that triggers neutrophil chemotaxis and degranulation [36], while monocyte chemoattractant protein-3 (MCP-3/CCL7) cleaved by MMP-2 acts as an antagonist that dampens inflammation [37]. Notably, MMP2 was downregulated while CCL7 was upregulated in our infection model (see below). Matrix metalloproteases (MMP9 in particular) in both macrophages and neutrophils have been implicated as proangiogenic factors [38] and have a role in cell migration and extracellular matrix degradation [39]. In experimental visceral leishmaniasis, high levels of MMP-9 have been associated with the disorganization of white pulp of the spleen [30]. Moreover, it was postulated that aberrant MMP-9 levels in the serum of dogs with canine leishmaniasis (CanL) would contribute to the development of splenomegaly [39]. Taken together, these findings may account for the upregulation of MMP genes as one of the factors involved in visceralization. Nevertheless, caution should be taken since the *in vitro* infection is a simplified model in which the response is limited to macrophage-like infected cells.

Other upregulated genes included SMAD7, which negatively regulates TGF-β1 signaling and mediates the interaction between TGF-β1 and other signaling pathways; the interferon-induced ISG20, which has 3'-5'-RNA exonuclease activity; PTGS2, a key enzyme in prostaglandin biosynthesis, which was found upregulated in macrophages infected with the closely related species *L. donovani* [40]. Importantly, the genes coding for chemokines CXCL2, CXCL3, CXCL5, CXCL6, CCL3, CCL4, CCL4L2, CCL3L3, CCL7 (involved in leukocyte migration) were also found upregulated. Interestingly, CCL3 and CCL4 were found upregulated in a *L. infantum*-infected murine model [41]. In particular, CCL3 has been shown to be overexpressed in the spleen during VL (42) and it has been included in a linear multivariant regression model to predict the parasitic burden [41].

On the other hand, among the 545 downregulated genes in *L. infantum* infected cells were MMP2, MMP15, MMP25, IFNGR2 (which is necessary for the IFN-γ-specific activation of the intracellular signal transduction pathway), IL16 and IL10RB (IL10 receptor subunit beta) that, together with downregulation of IL10RA observed in all infected cells, could impair the IL-10-mediated anti-inflammatory activity. Interestingly, the CYBB/NOX2 gene (encoding a critical component of the membrane-bound NADPH oxidase) was also downregulated, probably contributing to the mitigation of ROS production in the early infection.

At 48h post-infection, among the 60 genes upregulated exclusively in *L. infantum*-infected cells, it is worth mentioning BCL6 (an anti-apoptotic and anti-inflammatory gene), and the chemokine receptor CCR5, whose expression characterizes monocytes that are highly susceptible to be infected by *Leishmania* [43,44]. The CCR5 induction, as well as the upregulation of IL1B and PTGS2, has been previously reported in a *L. infantum*-infected murine model by Ontoria et al., who proposed that induction of CCR5 allows the parasites to enter silently into the macrophage and successfully establish inside the host [41]. Other genes included: CCN3, which plays a role in various cellular processes including proliferation, adhesion, migration, differentiation, and survival; ATG9B, involved in autophagy by mediating autophagosomal membrane expansion; APOA4, which was found upregulated in all infection conditions at 24h, and was still upregulated at 48h only in *L. infantum*-infected cells.

Among the 153 genes downregulated in *L. infantum*-infected cells, MMP2 and IL16 were still downregulated, as seen at 24h post-infection, while the chemokine CCL23 was downregulated only at 48h post-infection.

## Regulation of polarization in macrophage-like cells infected by *L. infantum*

Polarized macrophages can be broadly divided into two phenotypes: M1 (classically activated, pro-inflammatory subtype exhibiting microbicidal properties) and M2 (alternatively activated,

anti-inflammatory/regulatory subtype) [45]. Macrophage subtypes differ in the expression and production of cytokines, receptors, surface molecules, and transcription factors. For instance, M1 macrophages are characterized by the production of pro-inflammatory cytokines (e.g., TNF, IL-1β, IL-6, IL-12, IL-18, IL-23, and type I interferon). The subtype M2 can be further subdivided into M2a (polarized by IL-4 or IL-13), M2b (polarized by immunocomplexes, IL-1R and TLR ligands), M2c (polarized by IL-10, TGF-β or glucocorticoids) and M2d (induced by IL-6, TLR ligands and adenosine A2A receptor agonists) [46]. This phenotype is involved in inhibiting the immune response and promoting angiogenesis [47]. A different macrophage phenotype showing anti-oxidant potential called MOX, regulated by NFE2L2/NRF2 and having a role in chronic inflammation, has also been identified [48]. In the context of *Leishmania* infection, recent studies suggest that a balance between the initial microbicidal response of M1 macrophages and the regulatory functions exerted by M2 macrophages may be the key to healing [46].

In our model, a clear polarization of macrophages cannot be evinced by dysregulated genes (see also below); rather, a mixed polarization/activation phenotype can be argued. Nevertheless, a certain number of transcripts considered M1-associated [42] and MOX-associated [48] resulted particularly enriched in *L. infantum*-infected cells at 24 h compared with cells infected by *L. major* or *L. tropica* (Table 3), suggesting a peculiar response induced by *L. infantum* infection. Notably, the upregulation of several of those genes has been previously reported also in experimental VL models [41,42].

## Identification of common cellular processes involved in *Leishmania* infection

To identify known cellular processes shared by cells infected by three *Leishmania* species, we performed a pathway enrichment analysis using the upregulated and downregulated genes identified in *L. infantum*, *L. major* and *L. tropica*-infected cells as described in methods. The complete set of enriched terms for up- and downregulated genes (at both time post-infection) is listed in S5 and S6 Tables, respectively. Overall, the involvement of a broad repertoire of immune-related pathways both in the upregulated and the downregulated genes reflects the complexity and the importance of the balance between different type of response (e.g., Th1, Th2) occurring upon *Leishmania* infection, extensively reviewed in Rossi et al. [49].

At 24h post-infection, two multiple lists containing 749, 401 and 706 upregulated genes and 864, 500 and 733 downregulated genes for *L. infantum*, *L. major* and *L. tropica*, respectively, have been analyzed. Many of the pathways that were commonly enriched in the upregulated set of genes are related to immune signaling or infection (e.g., signaling pathways related to the response to interferons and other cytokines, NF-kB signaling, NOD-like receptor signaling pathway, unfolded protein response, adaptive immune system -including MHC class I mediated antigen processing & presentation-, RIG-I-like receptor pathway, lipid and atherosclerosis -including Toll-like receptor signaling pathway-) (Fig 4A). We previously reported the induction of Unfolded protein response (UPR) in macrophages differentiated from human monocytic cell lines (U937 and THP-1) and murine primary macrophages infected by *L. infantum* [50]. Moreover, the role of UPR during *Leishmania* infection has been evidenced also in RAW 264.7 cells infected with *L. amazonensis* [51]. Here we confirm and extend these findings to *L. major* and *L. tropica* infected cells, supporting the UPR as a common cellular response to different *Leishmania* species and evidencing its deep interconnection with inflammation and immunity. Interestingly, Nonhomologous end joining and DNA repair were also among the terms significantly enriched in the upregulated set of genes. It has previously reported that *L. donovani* elicits the activation of the host DNA repair machinery in murine bone-marrow

**Table 3. Expression values of selected M1-, M2- and MOX-associated genes significantly dysregulated in infected cells 24 h post-infection.**

| Gene symbol | *L. infantum*-infected cells (log2 fold change) | *L. major*-infected cells (log2 fold change) | *L. tropica*-infected cells (log2 fold change) |
|---|---|---|---|
| M1-associated genes | | | |
| CCL3 | 1.45 | ns | -1.36 |
| CCL4 | 2.61 | ns | -2.67 |
| CXCL2 | 3.51 | ns | ns |
| CXCL3 | 7.56 | ns | ns |
| CXCL5 | 6.42 | ns | ns |
| IL1B | 2.86 | ns | ns |
| IL6 | 4.33 | ns | ns |
| SOCS3 | 1.41 | ns | ns |
| PTGS2 | 3.12 | ns | ns |
| STAT1 | 2.34 | 2.52 | 2.33 |
| TNF | ns | ns | -1.51 |
| M2-associated genes | | | |
| IL1RN | 3.61 | 1.95 | 1.72 |
| CD226 | 4.13 | 3.65 | 3.74 |
| CD163 | ns | -1.88 | -2.00 |
| MMP9 | 1.25 | ns | ns |
| MOX-associated genes | | | |
| BTG1 | ns | 1.34 | 1.16 |
| DUSP1 | 1.90 | ns | ns |
| FEM1C | 1.58 | 1.18 | 1.30 |
| GCLC | 1.61 | ns | ns |
| GCLM | 2.82 | ns | ns |
| HMOX1 | 4.67 | 2.31 | 2.40 |
| KLF9 | ns | ns | 1.35 |
| RIT1 | 1.03 | ns | ns |
| SAMD8 | 1.03 | ns | ns |
| SLC16A6 | 2.71 | ns | ns |
| SRXN1 | 1.75 | ns | ns |
| TXNRD1 | 2.49 | ns | ns |

ns: not significantly dysregulated

macrophages [52]. Here we extend this finding to human-derived U937 macrophages infected with *L. infantum*, *L. major* and *L. tropica*, evidencing this host cell response as a common mechanism elicited by *Leishmania* infection, probably to prevent oxidative-driven macrophage apoptosis at early stages of infection. However, further studies are necessary to fully understand the molecular basis of activation of the host DNA repair pathways by the parasite. The common terms enriched in the downregulated genes (Fig 4B) are mainly associated with metabolism of lipids (cholesterol metabolism, SREBF and mir33 in cholesterol and lipid homeostasis, metabolism of lipids) but also with immune-related process (cytokine signaling, Neutrophil degranulation, Adaptive Immune System, MHC class II antigen presentation, phagosome, lysosome).

After 48h from infection, the number of dysregulated genes is largely reduced, along with the terms significantly enriched. Two multiple lists containing 124, 116 and 54 upregulated

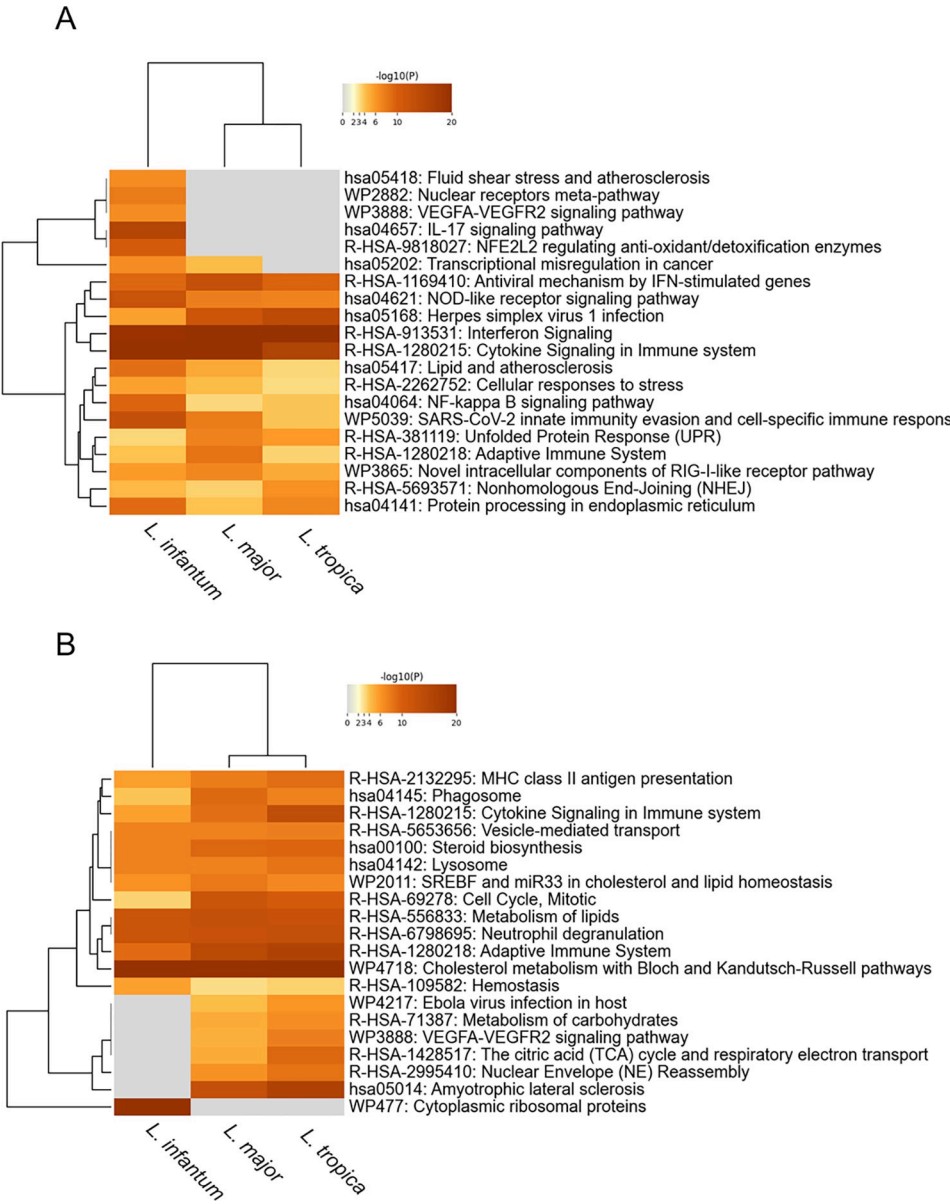

**Fig 4. Metascape meta-analysis results based on multiple genes dysregulated at 24 h post-infection.** The enrichment analysis was carried out with KEGG Pathway, Reactome Gene Sets, and WikiPathways as ontology sources. Heatmap showing the top 20 enrichment clusters for upregulated (A) and downregulated (B) set of genes found in *L. infantum*-, *L. major*- and *L. tropica*-infected cells. Significant functional enrichment terms were defined as those with p< 0.001. Gray color indicates a lack of significance.

genes and 109, 243 and 49 downregulated genes for *L. infantum*, *L. major* and *L. tropica*, respectively, have been analyzed. Among the commonly enriched terms in the upregulated set of genes, the terms related to interferon and response to virus and bacterium are maintained, while other terms appeared such as nicotinate metabolism (linked to the upregulation of $NAD^+$-consuming enzymes PARP9 and PARP14) or nucleic acid metabolism and innate immune sensing (Fig 5A). The common terms enriched in the downregulated genes (Fig 5B) are still associated with metabolism (cholesterol metabolism, metabolism of lipids) and with

A

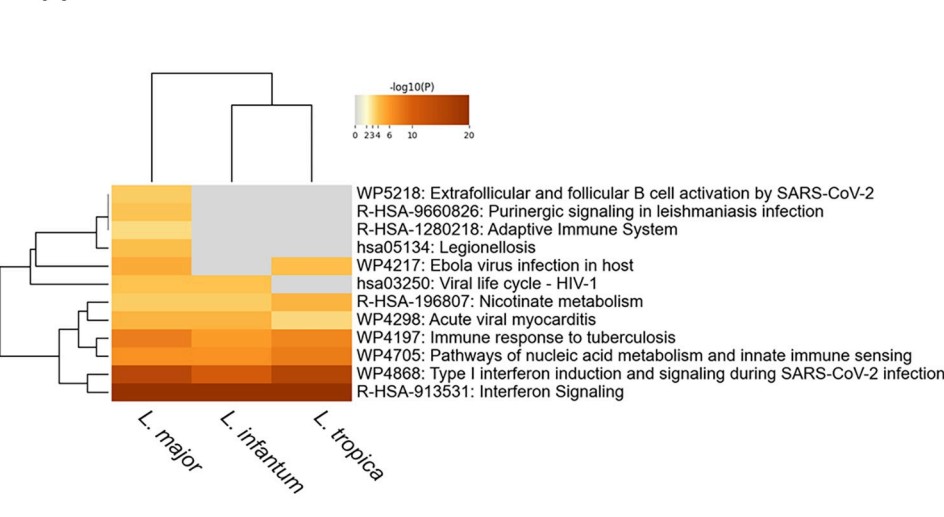

B

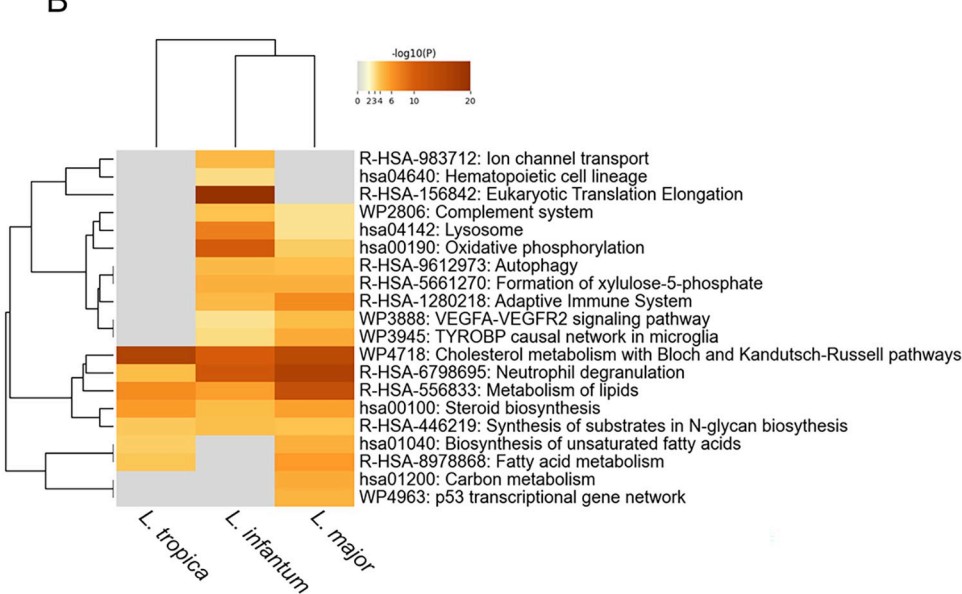

**Fig 5. Metascape meta-analysis results based on multiple genes dysregulated at 48 h post-infection.** The enrichment analysis was carried out with KEGG Pathway, Reactome Gene Sets, and WikiPathways as ontology sources. Heatmap showing the top 20 enrichment clusters for upregulated (A) and downregulated (B) set of genes found in *L. infantum*-, *L. major*- and *L. tropica*-infected cells. Significant functional enrichment terms were defined as those with p< 0.001. Gray color indicates a lack of significance.

immune-related process (Neutrophil degranulation), confirming -albeit attenuated- the findings at 24h.

Interestingly, the most significant terms enriched in the downregulated set of genes shared among all infected cells, both at 24 h and 48h post-infection, were related to cholesterol biosynthesis and lipid metabolism. A similar result was obtained by microarray analysis in *L. (Viannia) braziliensis*-infected U937-derived macrophages, in which a functional enrichment analysis allowed to identify the down-regulation of the steroid and sterol/cholesterol biosynthetic processes at 72 h post-infection [5]. Moreover, other studies performed on BALB/c

macrophages infected with *L. major* evidenced decreased cholesterol levels 72 h post-infection through the down-regulation of the gene HMGCR [53,54]. The reduction of the cellular cholesterol level could lead to enhanced membrane fluidity, disruption of rafts, and impaired antigen-presentation to the T cells. On the other hand, it has been shown that free cholesterol accumulation characterized the initial stages (up to 24h) of infection [55]. More recently, a study on *L. infantum*-infected THP1 cells reported an increase in the cholesterol content of the THP1 cell plasma membranes at 96 h post-infection [56]. In summary, the changes in cholesterol levels following internalization of *Leishmania* into the macrophage remain controversial, depending on time of infection, the infection model and parasite species, as well as from the balance between cholesterol metabolism, uptake, efflux, and storage. In our experimental model, cells infected by *L. infantum*, *L. tropica* and *L. major* showed the same downregulation of cholesterol biosynthesis and lipid metabolism, accounting for a common mechanism to establish infection, regardless the pathogenesis of single species. The cholesterol and fatty acid biosynthesis are regulated by transcription factor sterol regulatory element-binding proteins (SREBPs) and mTOR complex 1 [57]. Notably, the functional enrichment analysis evidenced SREBP-related terms (i.e., Activation of gene expression by SREBF; Regulation of cholesterol biosynthesis by SREBP; SREBP signaling; SREBF and miR33 in cholesterol and lipid homeostasis) in the downregulated set of genes either at 24h and 48h post-infection (S6 Table). Alterations in host cell lipid metabolic pathways have been reported in previous studies on murine models, highlighting the close relationship among host lipid metabolism, immune response, and the prognosis of the disease [58,59].

## Identification of cellular process elicited by species-specific expression signatures

At 24h post-infection, the results of pathway enrichment analysis with the upregulated multiple gene list, (Fig 4A) evidenced that pathways enriched only in *L. infantum*-infected cells included: VEGFA-VEGFR2 signaling pathway, NFE2L2 regulating anti-oxidant/detoxification enzymes, IL17 signaling (including TNF signaling, IL18 signaling and Interleukin-4 and Interleukin-13 signaling). Moreover, Nuclear receptors meta-pathway and Fluid shear stress and atherosclerosis (comprising Aryl hydrocarbon receptor pathway, that mediates inflammation and production of TNF in the context of *Leishmania* infection) [60] were also included.

The VEGFA-VEGFR2 signaling pathway activates angiogenesis by inducing cell proliferation, survival, and migration. Notably, vascular remodeling, including angiogenesis and lymphangiogenesis, is a feature of inflammatory microenvironments, and it occurs in both visceral and cutaneous leishmaniasis [61,62]. Moreover, lymphangiogenesis has been demonstrated to be critical for lesion resolution in *L. major*-infected C57BL/6 mice [63], a mouse strain that tends to be resistant to *Leishmania* infections.

The activation of the NRF2/NFE2L2 pathway leads to the reduction of cellular oxidative stress and is associated with the Th2-related induction of immune cells [64]. NRF2/NFE2L2 also regulates the MOX phenotype, which has a role in chronic inflammation [48]. Interestingly, several MOX-associated genes were found upregulated in *L. infantum*-infected cells (Table 3), accounting for the reliability of functional enrichment analysis. Moreover, in murine infection models, the canonical signature of the NRF2/NFE2L2 pathway has been reported in *L. amazonensis*- and *L. donovani*-infected macrophages, while *L. major* does not appear to use this pathway to subvert host cell defenses [64,65]. We confirmed these findings since NFE2L2-related pathway appeared downregulated in *L. major*-infected cells (see below) while it was significantly upregulated in cells infected by *L. infantum*, a species closely related to *L. donovani*.

IL17 signaling is a pro-inflammatory pathway that induces the production of several cytokines (e.g., IL1B, IL6, TNF), metalloproteases and chemokines, modulating immune cell trafficking. The Th17 subsets of T cells contribute to a protective immune response against VL. This protection is mainly mediated by the secretion of IL-17 and neutrophil infiltration [66,67]. At the early phase of VL infection, a strong IL-17 response was observed, followed by a progressive reduction to basal level during chronic VL, due to suppression by Treg lymphocytes [68]. Notably, TNF, IL-1, IL-13, IL-17, and IL-18 are among the cytokines involved in the host defense during VL [69]. In fact, in human visceral leishmaniasis IL-17, TNF and IL-18 have been associated with protection [69,70]. Moreover, IL4 and IL13 were found to be upregulated in VL patients but may have a dual role (disease progression/host protection) in leishmaniasis, depending on *Leishmania* species or host [69]. In our experimental model, the pathways involving these cytokines were clearly upregulated only in *L. infantum*-infected cells, accounting for a specific response to this viscerotropic species.

Concerning the downregulated genes, the results of pathway enrichment analysis (Fig 4B) evidenced that the main term enriched exclusively in *L. infantum*-infected cells was cytoplasmic ribosomal proteins (including terms such as Peptide chain elongation, metabolism of aminoacids, ribosomes). Interestingly, the enriched pathways shared between *L. major* and *L. tropica*-infected cells included: VEGFA-VEGFR2 signaling pathway, which was upregulated in *L. infantum*-infected cells; TCA cycle and respiratory electron transport, which has been shown to be inhibited also in murine macrophages infected by *L. major* [55]; metabolism of carbohydrates and Amyotrophic lateral sclerosis, which included a number of terms such as apoptosis, G2/M and G1/S transition, KEAP1-NFE2L2 pathway, noncanonical NFkB signaling, Parkin-ubiquitin proteasomal system pathway, Oxygen-dependent proline hydroxylation of Hypoxia-inducible Factor Alpha, cellular response to hypoxia, interleukin-1 signaling (S6 Table). Interestingly, the VEGFA-VEGFR2 and NFE2L2-related pathways were significantly enriched in the upregulated genes in *L. infantum*-infected cells and downregulated genes shared between *L. major* and *L. tropica*-infected cells (Fig 4) (S6 Table). Therefore, viscerotropic and dermotropic species appear to induce opposite responses regarding these two pathways in our infection model at the early time of infection. It is worth mentioning that increasing the number of *Leishmania* strains tested could strengthen these findings. Moreover, additional information regarding the importance of host or pathogen determinants for visceralization could be obtained including also *L. infantum* dermotropic strains circulating in Mediterranean region.

After 48 h from infection, a difference between viscerotropic and dermotropic species was much less evident. In fact, considering the upregulated genes, there were no terms exclusively enriched in *L. infantum*-infected cells. However, the terms purinergic signaling in leishmaniasis, adaptive immune system, and Extrafollicular and follicular B cell activation by SARS-CoV-2, were enriched exclusively in *L. major*-infected cells (Fig 5A). Concerning the enriched terms derived from downregulated genes, hematopoietic cell lineage, translation elongation, and ion channel transport were enriched in *L. infantum*-infected cells while fatty acids metabolism and biosynthesis of unsaturated fatty acids were shared between *L. major*- and *L. tropica*-infected cells. Interestingly, some terms were now shared between *L. infantum* and *L. major*-infected cells (e.g., adaptive immune system, VEGFA-VEGFR2 signaling pathway and oxidative phosphorylation).

## Conclusions

In this article, we investigated by RNA-Seq the transcriptional responses in U937-derived macrophages infected by different *Leishmania* species endemic in the Mediterranean basin: *L.*

*infantum*, *L. major* and *L. tropica*. Aside from a common transcriptional response eliciting the upregulation or downregulation of several cellular processes, the different *Leishmania* species also triggered a significantly different response in human macrophages, with several functional pathways appearing activated or deactivated in response to different species. In fact, U937-derived macrophages appear to discriminate between *L. infantum*, *L. major* or *L. tropica* at early infection in terms of the number of dysregulated genes and functional pathways activated or repressed. Among relevant activated functional pathways in *L. infantum* infected cells are the VEGFA-VEGFR2 signaling pathway, NFE2L2 regulating anti-oxidant/detoxification enzymes, and IL17 signaling. Among the downregulated pathways shared between *L. major* and *L. tropica*-infected cells are TCA cycle and respiratory electron transport, and metabolism of carbohydrates. These pathways could be considered to better characterize pathogenetic mechanisms in species able to cause visceral or cutaneous disease and identify targets for a host-directed therapeutic strategy that can be used in combination with canonical anti-leishmanial compounds.

Nevertheless, it is worth mentioning that the infection model used in this work consists of an *in vitro* differentiated tumor cell line which, although similar, may not perfectly mimic the behavior of a human macrophage and it is certainly a simplified model respect to *in vivo* infection, thus representing a limitation of this approach.

## Supporting information

**S1 Fig. Infection index in U937-derived macrophages infected with *L. infantum* (MHOM/IT/08/31U), *L. major* (MHOM/SU/73/ASKH) and *L. tropica* (MHOM/SU/74/K27) at 24h and 48h post-infection.**
(PDF)

**S2 Fig. Evaluation of selected targets at protein levels.** A) Representative western blot showing the evaluation of DDIT3, ATG7, ERO1-L-α (ERO1A), NQO1, HSP70 family protein 5 (HSPA5), and c-Myc (MYC) proteins in U937-derived macrophages infected with *L. infantum*, *L. major*, *and L. tropica* for 24h and 48h. Actin was used as loading control. The protein levels were analyzed in total cell lysates, and band density quantification was performed using a Chemi-Doc System. Densitometry values for specific proteins normalized against non-infected cells are included above each lane. B) IL-1β normalized to non-infected cells measured in supernatant of *L. infantum*-infected cells through ELISA test; C) Gelatin zymography of lysates from U937 cells are characterized by gelatinase's forms belonging to the MMP-9 class; all gelatinolytic bands appear more evident in lysates from *L. infantum*-infected cells, in particular at 48h post-infection. D) Correlation analysis of Log2 fold change of mRNAs (determined by RNA-seq) and the corresponding proteins (determined by western blot, ELISA or gelatin zymography), at 24h post-infection (Spearman's rho, $\rho = 0.87$, p<0.001), and 48h post-infection (Spearman's rho, $\rho = 0.08$, p = 0.7).
(TIF)

**S1 Table. The complete lists of genes significantly deregulated after 24h infection in U937-derived macrophages infected with *L. infantum*, *L. major*, and *L. tropica*.** Genes with a log2 fold change >1 and <-1, and FDR-adjusted p-values < 0.05 are included.
(XLSX)

**S2 Table. The complete lists of genes significantly deregulated after 48h infection in U937-derived macrophages infected with *L. infantum*, *L. major*, and *L. tropica*.** Genes with a log2 fold change >1 and <-1, and FDR-adjusted p-values < 0.05 are included.
(XLSX)

**S3 Table. Genes consistently up- and downregulated across 24h and 48h in U937-derived macrophages upon infection with *L. infantum*, *L. major* and *L. tropica*.**
(XLSX)

**S4 Table. List of significantly deregulated genes shared by U937-derived macrophages infected with different *Leishmania* species and deregulated genes found exclusively in *L. infantum*, *L. major* or *L. tropica*-infected cells.** Upregulated and downregulated genes at 24h and 48h post-infection are listed in different excel spreadsheets.
(XLSX)

**S5 Table. The complete set of enriched terms for upregulated genes at 24h and 48h post-infection, obtained after pathway enrichment analysis using the multiple gene list tool in Metascape.**
(XLSX)

**S6 Table. The complete set of enriched terms for downregulated genes at 24h and 48h post-infection, obtained after pathway enrichment analysis using the multiple gene list tool in Metascape.**
(XLSX)

## Author Contributions

**Conceptualization:** Fabrizio Vitale, Mauro Magnani, Luca Galluzzi.

**Formal analysis:** Aurora Diotallevi, Giuseppe Persico, Luca Galluzzi.

**Funding acquisition:** Fabrizio Vitale, Mauro Magnani.

**Investigation:** Aurora Diotallevi, Gloria Buffi, Marcello Ceccarelli, Daniela Ligi, Ferdinando Mannello.

**Methodology:** Aurora Diotallevi, Federica Bruno, Germano Castelli, Gloria Buffi, Marcello Ceccarelli, Daniela Ligi, Ferdinando Mannello.

**Writing – original draft:** Aurora Diotallevi, Luca Galluzzi.

**Writing – review & editing:** Aurora Diotallevi, Federica Bruno, Germano Castelli, Giuseppe Persico, Luca Galluzzi.

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
