## [Decision Letter · Decision Letter 0]

16 Oct 2023

Dear Dr Galluzzi,

Thank you very much for submitting your manuscript "Transcriptional signatures in human macrophage-like cells infected by Leishmania infantum, Leishmania major and Leishmania tropica" for consideration at PLOS Neglected Tropical Diseases. As with all papers reviewed by the journal, your manuscript was reviewed by members of the editorial board and by several independent reviewers. In light of the reviews (below this email), we would like to invite the resubmission of a significantly-revised version that takes into account the reviewers' comments. 

We cannot make any decision about publication until we have seen the revised manuscript and your response to the reviewers' comments. Your revised manuscript is also likely to be sent to reviewers for further evaluation.

Sincerely,

Armando Jardim, PhD

Academic Editor

Ricardo Fujiwara

Section Editor

Reviewer's Responses to Questions

**Key Review Criteria Required for Acceptance?**

**Methods**

-Are the objectives of the study clearly articulated with a clear testable hypothesis stated?

-Is the study design appropriate to address the stated objectives?

-Is the population clearly described and appropriate for the hypothesis being tested?

-Is the sample size sufficient to ensure adequate power to address the hypothesis being tested?

-Were correct statistical analysis used to support conclusions?

-Are there concerns about ethical or regulatory requirements being met?

Reviewer #1: See all relevant comments "Summary and General Comments" below.

Reviewer #2: The Methods are well-described and include all necessary information. One missing data need be included: the software(s) used to make data analysis and graph construction.

Reviewer #3: Are the objectives of the study clearly articulated with a clear testable hypothesis stated? Generally yes.

-Is the study design appropriate to address the stated objectives? In overall design, but a little short on execution.

-Is the population clearly described and appropriate for the hypothesis being tested? Generally yes, but see next point.

-Is the sample size sufficient to ensure adequate power to address the hypothesis being tested? Regarding inferences they were making regarding differences between the clinically distinct species, the sample size was insufficient (see general comments section).

-Were correct statistical analysis used to support conclusions? The number of clinically distinct species was very low, with two representatives from one and only one from the other. This would explain why there was no statistical analysis for this major aspect of their study

-Are there concerns about ethical or regulatory requirements being met? No concerns.

**Results**

-Does the analysis presented match the analysis plan?

-Are the results clearly and completely presented?

-Are the figures (Tables, Images) of sufficient quality for clarity?

Reviewer #1: See all relevant comments "Summary and General Comments" below.

Reviewer #2: The results and discussion are clear and well-presented in the text.

Reviewer #3: -Does the analysis presented match the analysis plan? Yes 

-Are the results clearly and completely presented? Yes

-Are the figures (Tables, Images) of sufficient quality for clarity? Yes

**Conclusions**

-Are the conclusions supported by the data presented?

-Are the limitations of analysis clearly described?

-Do the authors discuss how these data can be helpful to advance our understanding of the topic under study?

-Is public health relevance addressed?

Reviewer #1: See all relevant comments "Summary and General Comments" below.

Reviewer #2: The conclusions are supported by the data. Limitations could be including, mainly about the type of cell used for the work.

Reviewer #3: -Are the conclusions supported by the data presented? The conclusions are overly speculative at times.

-Are the limitations of analysis clearly described? No

-Do the authors discuss how these data can be helpful to advance our understanding of the topic under study? Somewhat.

-Is public health relevance addressed? Yes

**Editorial and Data Presentation Modifications?**

Reviewer #1: See all relevant comments "Summary and General Comments" below.

Reviewer #2: I have only fewer comments.

• In line 234, use: genus Leishmania OR subgenus Leishmania (Leishmania);

• Figure 3: include the terms Upregulated and Downregulated above diagram columns (like 24h and 48h titles for lines)

• To make clearer, authors could carefully revise the gene and protein nomenclature, some terms (as in lines 300 and 317) are referring to proteins but graphed without hyphen.

• Line 421: use BALB/c

• The colors used in the Figure 4 are quite confusing. Could you change to two colors pattern (one for downregulated and other to upregulated genes)?

Reviewer #3: (No Response)

**Summary and General Comments**

Reviewer #1: In this manuscript, the authors carried out transcriptomic analyses on human macrophage-like cells (PMA-differentiated U937) infected with three Leishmania species, namely L. infantum, L. major, and L. tropica. Transcriptional signatures analyses revealed common but also differential host gene expression trends between the different parasite species. Furthermore, host gene expression was assessed at two different time points, offering a better appreciation of the kinetics of gene expression. This study adds incremental knowledge of the transcriptional perturbations caused by Leishmania parasites. Overall, the study design is straightforward, and analyses appear to be appropriately carried out. A few points should be addressed by the authors before accepting the manuscript for publication. 

1. Although the use of PMA to differentiate monocytic-like cell lines towards macrophage-like cells (ex: THP-1, U937) is standard practice, it is well recognized that PMA also transiently activates cells to produce inflammatory mediators (amongst other things). It would therefore be pertinent to mention this point in the discussion as it is undoubtedly a variable that influences gene expression and hence interpretation of the results. 

2. In the Methods section, the authors indicate that “Infections were repeated twice (one dish per infection)” (see line 120). Could the author clarify whether the infections were performed in two independent biological experiments, or if the replicates correspond to technical replicates (i.e., two separate plates but prepared as part of a single experiment)?

3. In the Results and Discussion section, the authors state that “[…] the trimmed reads mapped on the human were 44.6% and 66.1% at 24 and 48 h, respectively, while for the uninfected macrophages were 98.5% and 98.8% at 24 h and 48 h, respectively” (see lines 189-191). This raises a few questions:

• Were the mapped reads from samples infected from the different Leishmania species similar? It is not clear to which samples these values (i.e., 44.6% and 66.1%) correspond. Can the authors clarify?

• The values of mapped reads obtained from infected samples seem somewhat low, especially compared to uninfected samples. Presumably, this is due to “contaminating” reads coming from parasite-derived transcripts. Have the authors considered first filtering out reads that would map to the Leishmania genome prior to performing their sequence alignments on the human genome? In any case, can the authors elaborate on this point?

4. Although RT-qPCR analyses agree to a high degree and validate the transcriptomic analyses, the study would greatly benefit from the inclusion of a handful of validation at the protein level. Indeed, this would help show potential functional changes in the infected host cell, rather than simply at a mRNA level (which do not always correlate with biologically meaningful changes in protein expression and thus function). Have the authors monitor any of the regulated genes at the protein level?

Reviewer #2: Dear authors,

Firstly, congrats for the manuscript. It is a very interesting work and well-conduced. The manuscript writing has higher quality. The only missing part: I agree with you, the importance of make analysis in human macrophages, but these used cells are a lineage originated from cancer and differentiated using inflammatory activator. Please, you need clarify that these data could be a limitation of obtained results.

Reviewer #3: The distinct clinical manifestations of Leishmaniasis are largely associated with different Leishmania species. L. major and L. tropica, for example, are typically associated with cutaneous leishmaniasis, whereas species including L. infantum are usually associated with visceral leishmaniasis. The factors that determine these species-specific clinical differences are poorly understood. To address this medically important gap in knowledge, the authors examine macrophage transcriptional responses to Leishmania both common amongst, and distinct between these three Leishmania species.

Strengths:

The authors have identified macrophage transcriptional signatures common to infection with all three of these Leishmania species and also signatures specific to the two cutaneous species versus the visceral species.

Although comparing macrophage gene expression upon infection with a cutaneous vs. a visceralizing species is not entirely novel, this major aspect of their study nonetheless represents a new contribution to this important, but neglected area of research. 

Concerns:

These in vitro studies of the immortalized monocyte-like U937 line have somewhat limited physiologic relevance. Further, this study only focuses on the macrophage part of the response. It does not include other innate cells nor the adaptive response and other tissue specific aspects involved in the vivo response. The authors should mention these limitations.

It is unclear why so few transcripts were validated by RT-qPCR (apparently they only validated only 8 at the 24 hr timepoint). Instead of focusing on the few genes that were validated, the discussion of species-specific genes/pathways was focused on genes whose expression was not validated. This is a major disconnect that needs to be addressed experimentally, at their 24 and 48 hr timepoints

The authors emphasize that their study represents a systematic comparison of macrophages infected with Leishmania species that cause different disease outcomes, however to really be systematic and statistically valid, the work should include more representative species from each disease presentation. For example, why not at least include L. donovani, a visceralizing species that is also present in the Mediterranean, as well? Further, have the authors considered analyzing/discussing dermotropic strains of L. infantum that are also circulating in the Mediterranean region?

It would be more informative if the authors included a brief discussion of how/why some of their major findings may have differed from other studies of Leishmania species-specific effects on macrophage gene expression (e.g. Cox2).

Overall, the study is somewhat unfocused. Much of the time, they are just listing individual genes they identified, with standard descriptions of their reported functions. Although potentially interesting, without a functional follow-up, this does not contribute much insight into the determinants of clinically distinct forms of leishmaniasis. The authors might be better served keeping many of these genes in their supplementary data and instead focus their descriptions/discussion on the RT-qPCR-validated genes.

Minor points:

They used stationary phase promastigotes, ideally would be better to use purified metacyclics.

A phagocytosis control, such as latex beads as others have used, would have been more informative.

Line 333: “CCL23 was newly downregulated”; unclear what the authors mean here. Why not just say “down regulated”, as with the other 24hr p.i. genes, or is there something temporally different about CCL23?

Many minor issues with use of language, grammar and a few typos:

e.g.: Line 20: “Despite animal models and genomic/transcriptomic studies provided important insights, the pathogenic determinants….are still poorly understood” 

e.g. : Line 2-309: “ infected cells respect to” (should read “ infected cells with respect to”)

e.g. Lines 279-280 sound a bit odd: contrary to the authors assertions, genes don’t “unravel pathways”.

e.g. Line 381: what is a “multiple list”?

Summary

Overall, the authors should make clear that, whilst interesting and representing a contribution to knowledge in an important area, their findings represent a rather limited dataset (i.e. they present an in vitro study of the response of a single immortalized monocytic-like cell line to several Leishmania species, with a sample size insufficient to provide a meaningful stand-alone comparison between cutaneous and visceralizing Leishmania species). Importantly, the genes/pathways highlighted in their principal findings do not appear to have been validated. It is important that they validate these genes.

The authors should avoid over-interpretation/extrapolation. For example, one of their principal, yet unvalidated findings was that genes associated with the VEGF pathway were upregulated in L. infantum-infected cells. They interpreted this to indicate that vascular remodeling may be important for visceralization. Aside from the apparent lack of validation for VEGF genes, this interpretation remains overly speculative, especially since VEGF is known to also have roles outside of the endothelial/vascular system.

PLOS authors have the option to publish the peer review history of their article (what does this mean?). If published, this will include your full peer review and any attached files.

Reviewer #1: No

Reviewer #2: No

Reviewer #3: No
---

## [Editor Report · Decision Letter 1]

19 Mar 2024

Dear Dr Galluzzi,

We are pleased to inform you that your manuscript 'Transcriptional signatures in human macrophage-like cells infected by Leishmania infantum, Leishmania major and Leishmania tropica' has been provisionally accepted for publication in PLOS Neglected Tropical Diseases.

Best regards,

Armando Jardim, PhD

Academic Editor

Paul Brindley

Editor-n-Chief

---

## [Editor Report · Acceptance letter]

2 Apr 2024

Dear Dr Galluzzi,

We are delighted to inform you that your manuscript, "Transcriptional signatures in human macrophage-like cells infected by Leishmania infantum, Leishmania major and Leishmania tropica," has been formally accepted for publication in PLOS Neglected Tropical Diseases.

Best regards,

Shaden Kamhawi

co-Editor-in-Chief

Paul Brindley

co-Editor-in-Chief
